# Differentiated Protection and Hot/Cold-Aware Data Placement Policies through k-Means Clustering Analysis for 3D-NAND SSDs

Seungwoo Son and Jaeho Kim *

Department of Aerospace and Software Engineering & Engineering Research Institute,
Gyeongsang National University, Jinju 52828, Korea; ssw2608@gmail.com
* Correspondence: jaeho.kim@gnu.ac.kr

**Abstract:** 3D-NAND flash memory provides high capacity per unit area by stacking 2D-NAND cells having a planar structure. However, because of the nature of the lamination process, the frequency of error occurrence varies depending on each layer or physical cell location. This phenomenon becomes more pronounced as the number of flash memory write/erase (Program/Erasure) operations increases. Error correction code (ECC) is used for error correction in the majority of flash-based storage devices, such as SSDs (Solid State Drive). As this method provides a constant level of data protection for all-flash memory pages, there is a limitation in 3D-NAND flash memory, where the error rate varies depending on physical location. Consequently, in this paper, pages and layers with varying error rates are classified into clusters using the k-means machine-learning algorithm, and each cluster is assigned a different level of data protection strength. We classify pages and layers based on the number of error occurrences measured at the end of the endurance test, and for areas vulnerable to errors, it is shown as an example of providing differentiated data protection strength by adding parity data to the stripe. Furthermore, areas vulnerable to retention errors are identified based on retention error rates, and bit error rates are significantly reduced through our hot/cold-aware data placement policy. We show that the proposed differential data protection and hot/cold-aware data placement policies improve the reliability and lifespan of 3D-NAND flash memory compared with the existing ECC- or RAID-type data protection scheme.

**Keywords:** reliability; 3D-NAND flash; bit error rate; RAID; cluster; k-means





## 1. Introduction

As the demand for mass storage devices is rapidly increasing because of the recent development of big data and AI technology, 3D-NAND flash memory is establishing itself as a mainstream medium for storage devices such as SSDs. By stacking cells, 3D-NAND flash memory achieves high capacity per unit area, overcoming the integration limit of the existing 2D-NAND structure, which integrates cells in a planar manner. However, because of the nature of the lamination process, the frequency of error occurrence varies depending on each layer or physical cell location, and this phenomenon becomes more pronounced as the number of flash memory writes/erases (Program/Erasure (P/E)) increases [1].

In other words, SSDs employing 3D-NAND flash memory as a medium can provide high capacity per unit area; however, because of the high-density stacking structure, the frequency of errors may vary depending on the cell location [1–3]. Consequently, instead of conventional 2D, we show the error rate pattern based on the P/E cycle of 3D-NAND flash, and we introduce a method to improve reliability based on the characteristics of the 3D-NAND flash, where the frequency of errors can vary depending on each layer or cell location [4]. (This paper is an extension of the paper "Improve reliability of SSD through cluster analysis based on error rate of 3D-NAND flash memory and application

of differentiated protection policy" presented at the 64th Summer Conference of the KSCI in 2021.)

Most of SSDs adopt an error correction code (ECC) for error correction. This method provides a constant level of data protection for all-flash memory pages, so it is appropriate when the number of error occurrences for each page and layer is similar. However, as the number of P/E cycles increases, the number of errors in each layer and page varies significantly, so applying ECC to all pages is insufficient for dealing with the error characteristics of 3D-NAND flash memory.

In this paper, we classify the flash memory area according to the error rates at the end of the endurance test, where the number of errors for each layer and page differs significantly through the k-means machine learning clustering algorithm. For 3D-NAND SSDs, we use a differentiated RAID-like protection policy based on the bit error rate (BER). The strength of error protection is adjusted by assigning appropriate $n$ parities to error-prone regions and nonvulnerable regions. In Sections 4 and 5, the adequacy of the parity number $n$ is demonstrated and is evaluated using the BER analysis. We also propose a hot/cold-aware data placement policy for relieving retention error based on 3D-NAND process characteristics. The retention error rate of 3D-NAND flash memory varies depending on the physical location. It should be noted that the retention error rate based on physical location differs from the BER based on endurance. Consequently, it is critical to effectively deal with the retention error, which is one of the most serious errors in flash memories. We show that our two proposed policies significantly reduce the error rates compared with ECC and RAID, which are the existing error handling methods. For quantitative error rate comparison, we assume an SSD with ECC, RAID, and our proposed policies and analyze them using an analytical model. Among the existing studies considering the characteristic of flash memory, LI-RAID [5] and WARM [6] are the most similar to the two of our proposed policies, respectively. The differences from these previous studies are described in Sections 2.3 and 2.8

In this work, we make the following contributions:

- Differentiated Protection: Using the characteristic that the error rate of each page and layer of 3D-NAND flash memory is different, all blocks and pages are classified into clusters through the k-means algorithm. In clusters with high error rates, the number of parities is added to increase data protection strength, and in clusters with low error rates, the number of parities is reduced to set differentiated data protection strength.
- Hot/cold-aware data placement: It uses the characteristics of hot data, where data update and access attempts are active. Place hot data in the part vulnerable to retention error, and place cold data in the part less vulnerable to retention error.
- Analysis of Reliability of SSDs: Error rates of the ECC, RAID, and our proposed policies are compared and evaluated quantitatively by an analytical model.

The remainder of this paper is organized as follows. Section 2 discusses background and related work, especially 3D-NAND flash memory. Section 3 demonstrates our two proposed policies and describes the relationship between the two methods. Section 4 discusses analytic formulas used to analyze error rates. Section 5 shows a quantitative comparison of error rates between the proposed and existing methods. Section 6 concludes this work with a summary.

## 2. Background and Related Work

### 2.1. Flash Memory Features

NAND flash memory forms the basic building blocks of SSDs today, and most SSDs on the market connect a large number of NAND flash memories to a channel and parallelize them with an SSD controller to provide high performance and large capacity [7–11]. A NAND flash memory chip is made up of multiple dies and planes with multiple blocks, each with several fixed pages [7,9]. The most fundamental operations in flash memory are read and write operations, which are conducted in page increments. A typical page write

takes hundreds of microseconds, whereas a typical page read takes tens of microseconds, which is much faster than writing a page [12].

A unique feature of flash memory is that new data cannot be overwritten on a page once data has been written [13]. This is different from HDD where data can be overwritten. To write data back to the page where it was written, the block containing the page must first be erased. This erase operation is slower than the page write operation, and the number of erase-after-write operations, also known as the P/E cycle, is limited by the flash memory's manufacturing technology [14]. Four technologies are widely used today: single-level cell (SLC), multi-level cell (MLC), triple-level cell (TLC), and quad-level cell (QLC). Among them, TLC is widely used because it is inexpensive and suitable for large capacity [15].

### 2.2. Structure of 3D-NAND Flash Memory

Figure 1 shows the structure of 2D- and 3D-NAND flash memory [2]. Figure 1a shows the structure of 2D-NAND flash memory. It consists of a cell for storing data, a channel providing pathway for electrons transportation, and a drain and source providing a path for electron movement on the application of voltage to the cell. Conversely, 3D-NAND flash memory is made into a three-dimensional structure by stacking 2D-NAND flash memory, and one layer is called a layer. The blue cylindrical model in Figure 1b was created by vertically rotating the existing 2D-NAND flash memory. Storage capacity per unit space has increased significantly compared with 2D-NAND flash memory because of the three-dimensional design and process. Wordline, as shown in Figure 1b, is a passage through which the voltage applied to each cell passes. It can be seen that wordline and bit line are orthogonal to each other.

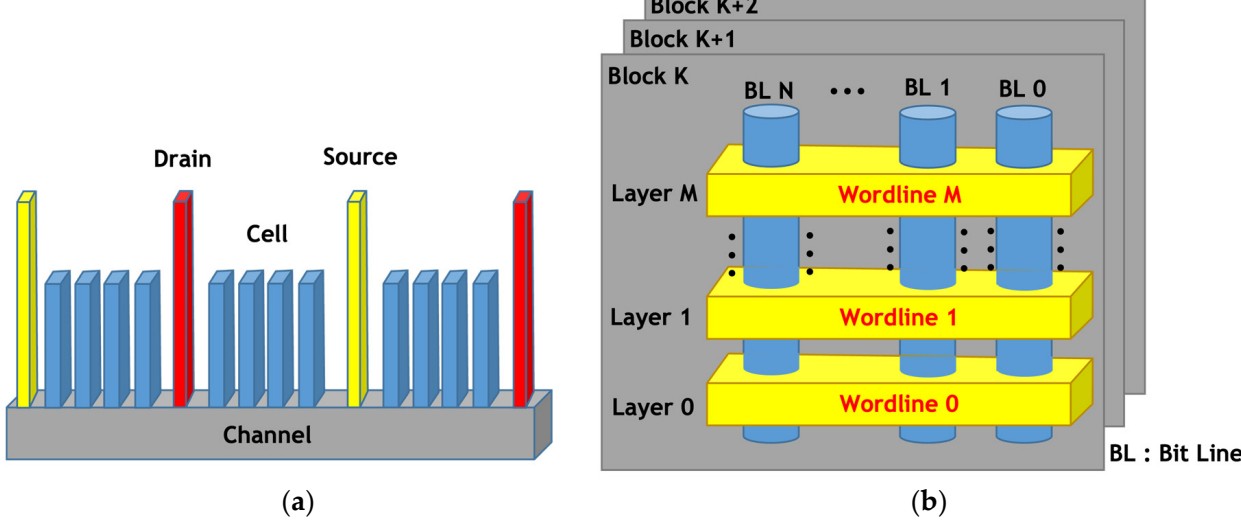

**Figure 1.** 2D (**a**) and 3D-NAND flash memory (**b**) structure [2].

One layer consists of upper, middle, and lower pages. Like 2D-NAND, 3D-NAND flash memory has several planes and dies, and one die consists of several blocks, and one block consists of several pages.

### 2.3. Reliability Issues Due to the Process-Variation of 3D-NAND Flash

3D-NAND flash memory is manufactured by stacking planar (2D) NAND cells. However, because of the nature of the stacking process, the frequency of errors may vary depending on each layer or cell location when the layers are stacked in this manner [1]. This problem can be influenced by factors such as structure distortion during processing, channel hole bending, and a change in the diameter of the channel hole from the upper layer to the lower layer. These process issues are representative factors that can affect flash memory reliability [2,3].

As a method to cope with 3D-NAND process variation, an error prediction method using machine learning [16] and LI-RAID [5] have been proposed. The error prediction method [16] predicts an error level through machine learning based on the raw bit error measured for each block and allocates a block group to be used according to the predicted level. LI-RAID [5] is the most similar to our proposed differentiated protection policy and it composes pages with different error rates into one stripe to keep the error rates of all stripes uniformly. The difference between LI-RAID and the proposed method is to adjust the protection strength by adjusting the number of parities. We believe that our proposed policy is a more flexible configuration.

### 2.4. Error Rate Pattern according to P/E Cycles of 3D-NAND Flash

Yuqian Pan et al., designed a NAND flash test platform with Xilinx Zynq-7020 and tested blocks on different chips of 3D-TLC flash memory [16]. Figure 2 depicts the distribution of error rates revealed in the papers by Yuqian Pan et al. [16]. In this figure, we show the average error rate as a function of block number, ranging from 50 to 5000 P/E cycles. The figure shows that the difference in average BER by block does not appear significantly below 1000 P/E cycles but as the number of P/E increases, the difference enlarges.

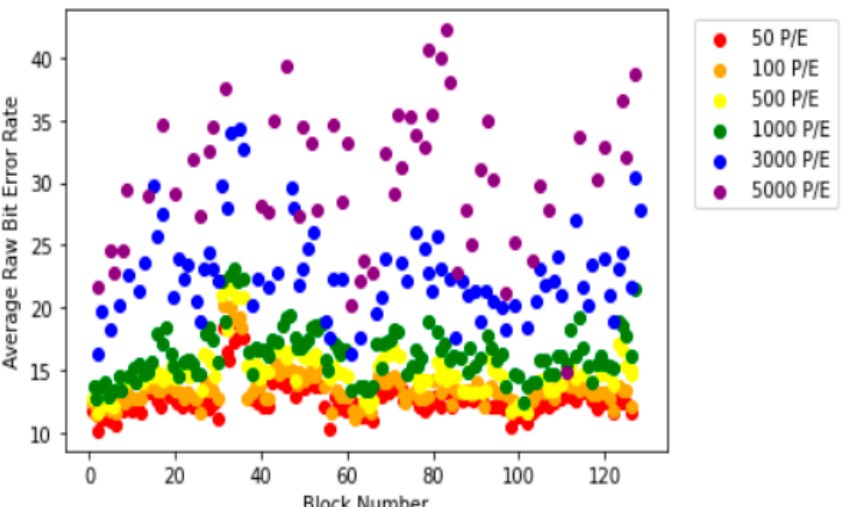

**Figure 2.** Average bit error rate per block number in different P/E cycles [16].

### 2.5. Number of Errors per Page and Layer

Zambelli et al., showed that the number of errors in 3D TLC NAND flash differs for each layer and page [1]. Figure 3 is a distribution chart of the number of errors reported by Zambelli et al. [1]. Endurance testing is a procedure that determines the number of incorrect bits for each page and layer after repeatedly writing and erasing a memory block in a random pattern. Figure 3 depicts an error bit distribution measured at the end of the endurance test reported by [1]. The *x*-axis layers in Figure 3 range from 0 to 1. The closer the layer is to 0, the lower the layer, and the closer the layer is to 1, the higher the layer. Furthermore, the Fail bits represent the number of times an error occurred, and it is expressed as a relative value ranging from 0 to 1. The black, blue, and red dots represent the top, middle, and bottom of the page, respectively. We can see the characteristics of the errors that occur in each layer in Figure 3. The lower page has the most errors in the lower layer, and the number of errors occurs regardless of the page type. Conversely, in the middle layer, you can see that the number of error occurrences for each page is comparable. In addition, in the upper layer, it can be seen that the upper page has the most number of errors and the lower page has the least number of errors.

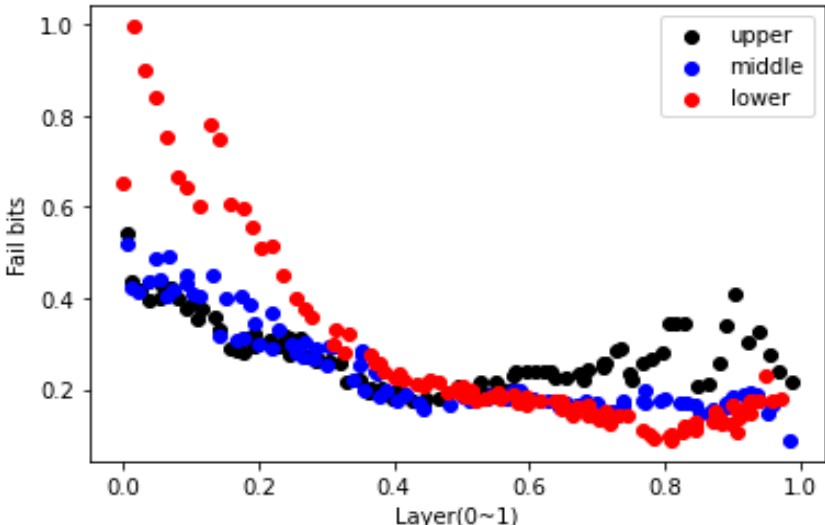

**Figure 3.** Number of error occurrences according to different layers and pages at the end of endurance test [1].

### 2.6. K-means Cluster Analysis Based on Error Rate

As shown in Figure 3, 3D-NAND flash has a characteristic that the number of errors is different for each page and layer. The reliability of 3D-NAND flash memory can be improved by applying a clustering algorithm to classify according to this characteristic and using a data protection policy suitable for the characteristics of each cluster.

There are various types of clustering algorithms, such as k-means, mean shift, Gaussian mixture model, and DBSCAN. Among them, the graph in Figure 3 was used as input data and classified using the k-means algorithm, which is easy and concise, and generally used frequently (see Figure 4) [17]. The *x*-axis in Figure 4 represents layers from 0 to 1. A value closer to 0 means the lower layer and a value closer to 1 means the upper layer. The *y*-axis indicates the number of error occurrences as Fail bits, and it is expressed as a relative value from 0 to 1 as in Figure 3.

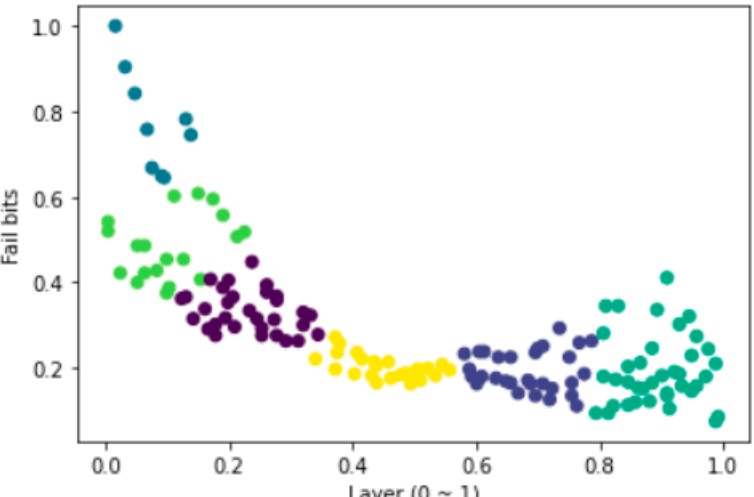

**Figure 4.** Number of error occurrences according to the layer that the k-means algorithm is applied.

In the k-means algorithm, the number of clusters is determined according to an arbitrary *k* value determined by the user, and similar clusters are classified only by the distance between the input data. The k-means algorithm seeks a set of data that minimize the variance of each cluster. Looking at how the k-means algorithm classifies clusters, the user determines the center point of the clusters as much as the *k* (the number of clusters)

value based on the first input data (here, the data in Figure 3). The data closest to each centroid are then classified into the same cluster using the Euclidean distance formula. The previously determined *k* center points are repositioned in the center of the newly classified cluster. Until the center point cannot be moved any further, the data closest to each center point are newly classified into the same cluster, and the process of moving the center point is repeated.

We adopt a heuristic way of increasing *k* by 1 to find an appropriate *k*. In this way, we obtain *k* values with clear error rate cluster identification. Result of classification, when the *k* value was 6 (the number of clusters was 6), areas according to the error rate were clearly identified.

*2.7. Write Amplification Factor (WAF)*

Another reason to use a data protection policy that is differentiated according to the characteristics of each cluster is related to the performance of the SSD. If our goal is simply to lower the error rate, we can apply the maximum data protection strength to all clusters. However, doing so will increase WAF because of indiscriminate parity data writing and adversely affect performance.

WAF stands for write amplification factor. WAF is the ratio of writes generated by an actual device (i.e., flash memories) to user write. In addition to simply writing data, the operation of finding and writing a new location in the flash memory for reading, updating, and rewriting data takes place. Consequently, even if the user attempts to write *n* bytes, *n* + *a* bytes are written inside the flash memory. WAF has a significant impact on flash memory storage I/O (Input and Output) performance; that is, as WAF increases, I/O performance decreases [12]. Parity data writes are another factor that increases WAF, which increases the number of P/E cycles, resulting in faster wear [12]. Rather than adding parity data to all clusters to reduce error rates, performance and reliability should provide differentiated data protection strength by selectively adding parity data based on the number of errors in each cluster.

*2.8. Retention Error*

A retention error is a data error caused by the loss of retained electric charge. When data is stored on a flash memory page and read after a certain period, the data value changes because of the loss of electric charge retained in the data. In terms of retention error, the error rate rises as retention time increases. The retention error tends to increase rapidly until about 11 days after data is written, after which the error increase rate tends to decrease [5]. Furthermore, because of the nature of the error, retention error occurs more frequently for cold data, where data update is difficult once it is stored on a page than for hot data, where data update occurs frequently. Consequently, by storing hot data in an area prone to retention errors, the error rate can be effectively reduced.

Figure 5 is the retention error distribution of 3D TLC NAND flash, which is reported by Zambelli et al. [1]. From the figure, it is obvious that the number of errors on the middle page is higher, whereas the number of errors on the lower page is relatively low. It can be seen that the number of errors on the upper page is less than on the middle page but greater than on the lower page. The overall distribution clearly indicates that the number of error occurrences is relatively high in the lower, middle, and 0.82–0.85 layers. Additionally, the number of errors in the upper layer is relatively low. Therefore, it can be seen that the areas vulnerable to retention errors are the middle page and the upper page of the lower, middle, and upper-middle layers.

There have been several studies [5,6,18–20] to reduce retention error, and the most similar technique to the proposed method is WARM [6]. WARM proposes a write-hotness aware retention management policy. It classifies hot and cold write data through a window-based algorithm to mitigate refresh overhead, which significantly increases the internal write amount of flash memory. WARM increases the lifespan of flash memory by selectively performing refresh only for write-cold. It exploits the property that write-hot data requires

a shorter retention time guarantee. Similar to WARM, our proposed hot/cold-aware data placement policy utilizes data hotness. The difference is that we exploit the error rate differentiation according to the physical location of 3D NAND to place hot- and cold-data at high and low retention error rates, respectively.

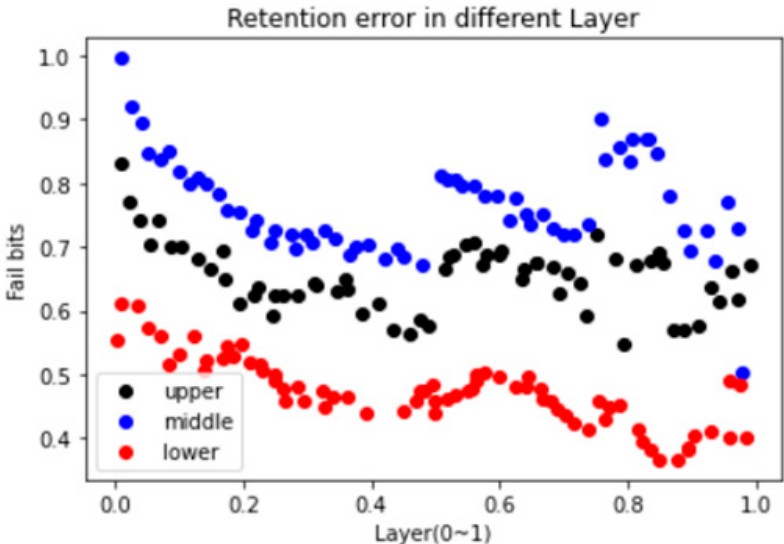

**Figure 5.** Number of retention error occurrences in different page and layer [1].

### 3. The Proposed Scheme

This section explains how to apply a differentiated protection policy to an SSD using 3D-NAND flash memory and a structure for arranging hot/cold data by identifying areas vulnerable to retention errors. Prior to the description, parity data is redundant data for error correction that is stored in the same stripe as user data when a page in a stripe is stored. When an error occurs in a specific piece of data in a stripe and the corresponding data cannot be read, the data in which the error occurred can be restored using the parity stored in the same stripe. Consequently, adding parity data can both lower the error rate and increase reliability. In Section 4, we investigated how much the error rate can be reduced by incorporating parity with analytic formulas.

#### 3.1. Differentiated Data Protection Policy Structure

First, Figure 6 briefly shows a general SSD internal structure to illustrate the proposed structure. It is assumed that five flash memory chips are connected by a channel to the SSD controller, and the block is configured as four pages. As shown in Figures 1 and 3, one layer has a lower, middle, and upper page, and the error rate varies by page and layer. Figure 4 depicts six clusters classified by the k-means algorithm based on the characteristics of these 3D-NAND flash memories to apply differentiated data protection strength based on error rates. If numbering the clusters in order from top left to right, depending on the error rates, the stripe of the first cluster sets the parity to $n$, the stripe of 2, 3, and 6 clusters sets the parity to $n-1$, and the stripe of 4 and 5 clusters sets the parity to $n-2$ to apply the differentiated data protection strength [14].

Assuming $n$ (by analyzing the error incidence analysis introduced in Section 4, the number of parity per stripe can be adjusted to achieve the desired reliability level) is 2, one stripe spans five flash memory chips. If flash memory blocks corresponding to the first cluster of the error occurrence are represented as shown in Figure 6c, it applies to the lower page of the lower layer. Similarly, the flash memory chip stripe corresponding to clusters 2, 3, and 6 can be expressed as Figure 6a,c, and it applies to all pages of the upper layer and to the upper and middle pages of the lower layer. If the flash memory chip stripe corresponding to the 4 and 5 clusters has a low number of error occurrences, it can be represented as Figure 6b and applies to all pages of the middle layer. The first cluster

that has the most number of error occurrences sets the data protection strength highest by setting the number of parity data per stripe ($n$ parity). For 2, 3, and 6 clusters, where the number of error occurrences is middle, the number of parity data per stripe is set to $n - 1$, and set the data protection strength of middle. Finally, for the 4 and 5 clusters that have the lowest number of error occurrences, the number of parity data per stripe is set to $n - 2$. So, specifies differentiated data protection strength for each cluster. The management methods of striping and partial stripe parity for partial data follow the recent efficient log structure writing schemes [12,14]. Differentiated protection policy should be similar to the efficient log write methods. Details will be discussed in Section 5.3.

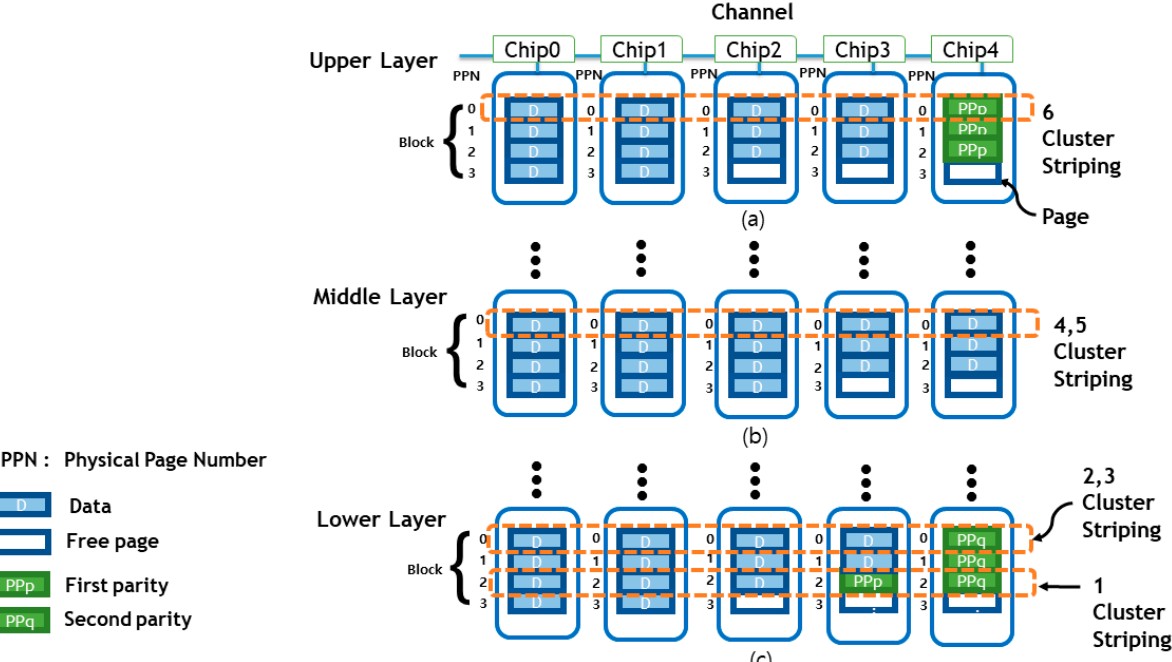

**Figure 6.** Example of differentiated data protection in SSD [12]. (**a**) Upper Layer; (**b**) Middle Layer; (**c**) Lower Layer.

### 3.2. Hot/Cold-Aware Data Placement Structure

As can be seen in Figure 5, retention error is also seen that the number of errors occurs differently by page and layer. It can be seen that the middle and upper pages have a relatively high number of errors, whereas the lower page has a relatively low number of errors. When looking at the number of errors by layer, it is clear that the lower layer has the highest number of errors. The middle and upper pages of the lower layer are vulnerable to the retention error, as are the middle and upper pages of the middle and upper layers, as well as the lower layer.

Retention error means an error that occurs when data is stored on the flash memory page and then read data after a certain period. This mostly happened with cold data, which is data that has been stored but has no access to it. Conversely, hot data is a relatively frequent update to the data in a short period after the data is stored on the page, and the probability of occurrence of retention error is low. If we arrange cold and hot data, we place hot data in a relatively vulnerable area to retention error.

In Figure 7, we present a data placement structure we proposed. We assume that cold and hot data exist at a ratio of 50% for the sake of simple notation. We also assume that the SSD is made up of five flash memory chips and that one block is made up of four pages, as shown in Figure 7. Let us start with the lower layer, which is the most vulnerable to the retention error. We use hot data striping on the middle page, which has the highest number of retention errors. Furthermore, if new hot data arrives, it can be placed on an empty upper page. Then, let us consider the middle layer, where the retention error rate

is lower than the lower layer but more than the upper layer. Similarly, we can see that hot data is organized on the middle page. However, since the retention error rate of the middle layer is lower than that of the lower layer, it can be seen that the number of hot data is relatively smaller than the lower layer. Finally, let us look at the upper layer that the number of errors is relatively the least shown. Hot data, like other layers, is placed first in the middle pages where the most errors occur. As the lower page of the upper layer is less vulnerable to the retention error, many cold data have been placed. Furthermore, because the number of errors is lower than in the middle layer, the number of hot data is lower. Additional overhead with the proposed policy will be discussed in Section 5.3.

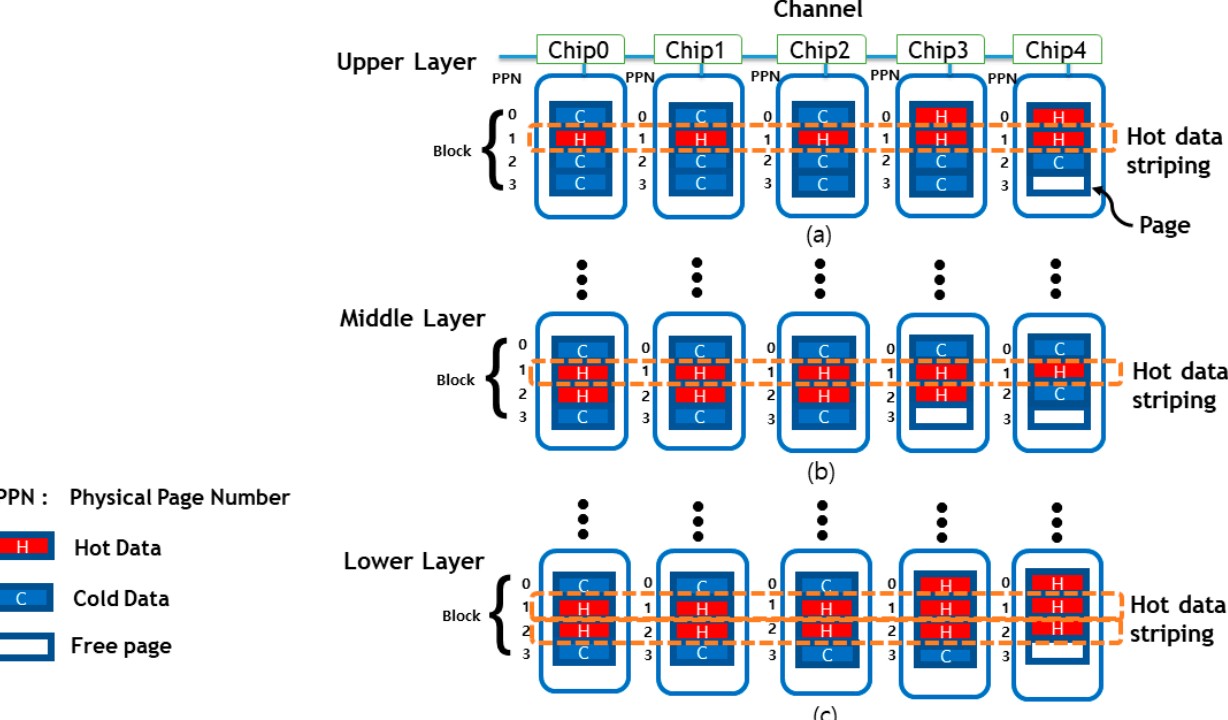

**Figure 7.** Example of data placement structure. (**a**) Upper Layer; (**b**) Middle Layer; (**c**) Lower Layer.

*3.3. Relationship between Two Policies*

The differentiated data protection policy and hot/cold-aware data placement technique presented in this paper can be independent and complementary data protection policies. The differentiated data protection policy can be applied flexibly and dynamically in response to changes in the error rate. For example, it can be selectively applied to the early and late stages of SSD by adjusting the data protection strength based on the change in the number of P/E cycles of the flash memory or depending on the bathtub model that appears in most systems. The hot/cold-aware data placement can consistently deal with retention errors throughout the SSD's lifetime.

## 4. Analysis of Reliability of SSDs

In this section, we explain analytic formulas used to analyze the ECC that protects data using only the existing ECC and RAID-like protection model that provides differentiated data protection strength (we also refer to this as Differentiated Protection in this work) in terms of error rates.

The raw bit error rate (*RBER*) of flash memory increases exponentially with the P/E cycle period as follows. In this equation, *x* represents the P/E cycle of the block, and *A* and *B* are constant values depending on the type and process of the flash memory [14].

$$RBER(x) = A \times e^{Bx} \tag{1}$$

ECC codes can correct errors up to *k* bits per *n*-specific code words. If ECC can correct *k* bits errors. The correctable page error rate *(CPER)* that ECC can correct from 0 bit error to *k* bits error rate, can be calculated with a binary distribution as follows [12,14].

$$CPER(n,k,x) = \sum_{i=0}^{k} \binom{n}{i} RBER(x)^i (1 - RBER(x))^{n-i} \tag{2}$$

Furthermore, there is a *k* + 1 or more bits error rate that ECC cannot correct, the uncorrectable page error rate *(UPER)* is as follow [12,14].

$$UPER(n,k,x) = 1 - CPER(n,k,x) \tag{3}$$

ECC has a limitation that ECC cannot correct errors when the number of error bits exceeds *k*, and errors that cannot be corrected by ECC are corrected through RAID parity. In addition, to correct the error through RAID parity, the error must first be detected through ECC.

ECC can correct errors up to *k* bits and detect errors up to 2 *k* bits. This is generally valid because coding theory supports this claim [12]. Additionally, assume that SSDs using RAID architecture can correct errors with parity only when a page has bit errors less than or equal to 2 *k*. Therefore, although it cannot be corrected, the detectable page error rate that can be detected by ECC is defined as follows [14].

$$DPER(n,k,x) = \sum_{i=k+1}^{2k} \binom{n}{i} RBER(x)^i * (1 - RBER(x))^{n-i} \tag{4}$$

There are two cases in which bit errors in the stripe can be corrected. The first case is when all pages containing a stripe have less than or equal to *k* bits errors. A case in which an error of *k* bits or less occurs in all pages containing a stripe can be corrected by ECC, and the probability of occurrence, in this case, is defined as follows [21].

$$\binom{N}{0} CPER(n,k,x)^N$$

The second case is when $N - 1$ pages of the stripe have *k* bits or fewer errors that occur, and one page has more than *k* bits and no more than 2 *k* bits errors occur. The probability of occurrence, in this case, is as follows [21].

$$\binom{N}{1} CPER(n,k,x)^{N-1} * DPER(n,k,x)$$

If there is no parity per stripe (only ECC is provided), the error bit will be corrected only by the ECC of the stripe page, so the *UPER* in the ECC model is as follows.

$$UPER_{ECC}(n,k,x) = \frac{1}{N}\left(1 - \binom{N}{0} CPER(n,k,x)^N\right) \tag{5}$$

*N* stands for stripe size, if there is one parity per stripe: (1) When only bit errors that can be corrected by ECC occur in the striped page, and (2) ECC detectable errors occur on one page of the stripe and ECC correctable errors occur on the remaining pages. Since errors can be corrected in these two cases, the *UPER* is as follows [14].

$$UPER_{P1}(n,k,x) = \frac{1}{N}(1 - \binom{N}{0} CPER(n,k,x)^N - \binom{N}{1} CPER(n,k,x)^{N-1} \\ * DPER(n,k,x)) \tag{6}$$

In the case where two parities per stripe are maintained, if two-page errors that can be detected by ECC occur at the same time, both pages can be modified at the same time, so the *UPER* is as follows [14].

$$
\begin{aligned}
UPER_{P2}(n,k,x) \quad &= \tfrac{1}{N}\left(1 - \begin{pmatrix} N \\ 0 \end{pmatrix} CPER(n,k,x)^N - \begin{pmatrix} N \\ 1 \end{pmatrix} CPER(n,k,x)^{N-1}\right. \\
&\left. *DPER(n,k,x) - \begin{pmatrix} N \\ 2 \end{pmatrix} CPER(n,k,x)^{N-2} * DPER(n,k,x)^2\right)
\end{aligned}
\tag{7}
$$

## 5. Evaluation

### 5.1. Apply Differentiated Data Protection Policy

Figure 8 shows the error rates of the four models for each cluster classified in Figure 4 based on Equations (1)–(6) in Section 4. In the figure, RBER represents the initial BER at 3000 P/E cycles, whereas ECC represents the BER of a conventional SSD that only uses ECC to protect data. Additionally, the model presented in this paper is Differentiated Protection. Differentiated Protection is a model that applies a differentiated data protection policy, such as the model presented in this paper. The BER is expressed by dividing the model into two parts: one without parity overhead and one with it. The initial RBER in Figure 8 was taken from the reference paper [22], and the error rate value indicated by each model in Figure 8 can be regarded as the average error rate value of each cluster. Furthermore, the black line near the error rate of $10^{-15}$ represents the industry standard error rate for a typical SSD [23]. The differentiated data protection policy without parity overhead model and ECC model are expressed using Equations (1)–(6), and the procedure is as follows.

1.  For RBER, the value measured in 3D TLC NAND flash memory [22] is used for calculating our analytical model.
2.  UPER is obtained through the ECC parameters of the reference paper [14] and Equations (2) and (4) of this paper. The UPER of Equation (4) obtained here is used to obtain the error rate of the ECC model without any parity data per stripe and the error rate of clusters 4 and 5 of the differentiated data protection policy.
3.  Clusters 2, 3, and 6 with one parity data per stripe of the differentiated data protection policy can receive data protection by one parity, so they are derived through Equations (3) and (5).
4.  Cluster 1 consists of two parities per stripe, and data protection can be obtained by two parities, so it can be calculated through Equation (6).

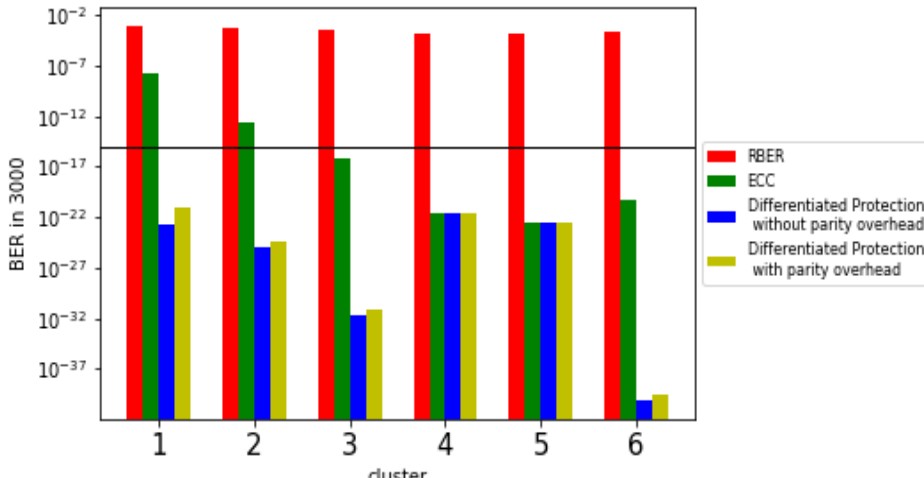

**Figure 8.** Bit error rate by cluster and model at the end of an endurance test.

We experimented with the financial workload on a 128 GB SSD in the work [14]. Through the experiment, we measure the amount of write including parity writes in

the SSD caused by the proposed policy. The amount of parity writes directly affects the P/E cycles and error rates of flash memory, so it is important to analyze SSD reliability. Compared with the case where the number of parity data per stripe is 0, the WAF increases by 20% when the number of parity data per stripe is 1 when the number of parity data per stripe is 2, the Differentiated Protection model considering parity overhead in Figure 8 is shown by referring to the result of 50% increase in WAF.

Looking at Figure 8, compared with the existing RAID model that uses only ECC to protect data, it can be seen that the model to which the differentiated data protection policy including parity overhead presented in this paper is applied shows the same or low error rate in all clusters. In clusters 4 and 5, it can be seen that the model that only uses ECC to protect data and the Differentiated Protection model have the same BER. As parity data are not added to clusters 4 and 5, there is no parity overhead in the Differentiated Protection model, and data is protected with the same protection strength as the model that protects data using only ECC.

### 5.2. Hot/Cold-Aware Data Placement Structure Applied

Figure 9a shows the BER comparison of the data placement structure by comparing the RBER aspect of retention error between the case without data placement and the case with data placement. Figure 9b shows the comparison of BER when ECC provided by default is applied to RBER. In addition, the RBER of Figure 9a was measured based on 9K P/E cycles of 3D TLC NAND flash memory [22]. Retention time, which is the *x*-axis of the graph, is a parameter indicating when a certain period has elapsed after data was saved, and was quantitatively evaluated by dividing it into four sections at two-week intervals from week 0 to week 6.

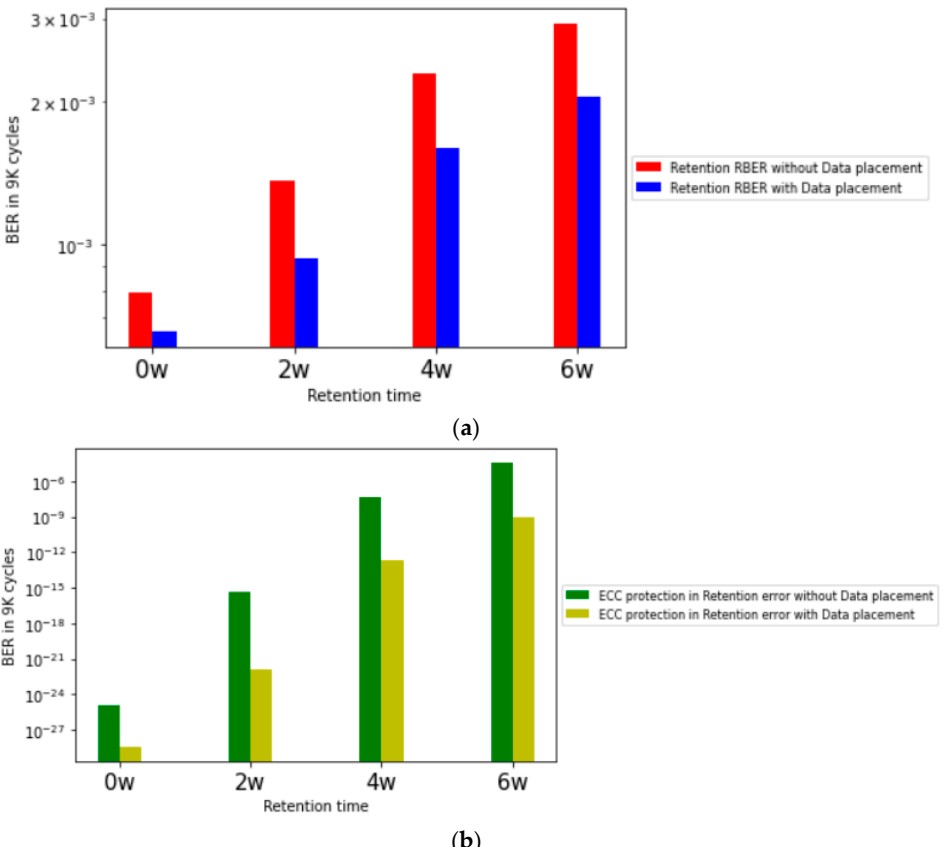

(a)

(b)

**Figure 9.** Retention error according to retention time with hot/cold-aware data placement model and without data placement model. (**a**) comparison of retention error between the case without data placement and the case with data placement. (**b**) comparison of BER when ECC provided by default is applied to RBER.

The red graph in Figure 9a shows the RBER of retention error for the existing data arrangement to which the data placement structure is not applied. The blue graph depicts the RBER of retention error when the data placement structure is used. As can be seen from the graph, data placement reduces RBER significantly, and the effect is more pronounced when the retention time is short. The reason for this is that as retention time increases, the likelihood of retention errors that occur also increases.

The green graph in Figure 9b shows the BER when ECC, the data protection policy provided by SSD, is applied based on the red graph in Figure 9a. Conversely, the yellow graph depicts the BER when ECC is used, based on the blue graph in Figure 9a. Even when ECC is used, it is clear that the data placement structure has a lower BER. As the probability of occurrence of retention error increases with retention time, the BER increases with retention time for both the structure in which the data is relocated and the structure in which the data is not relocated.

### 5.3. Discussion of Additional Cost

LI-RAID [5] and e-SAP [12] are the most similar to our proposed Differentiated Protection architecture. Therefore, the overhead of the proposed method is similar to these two methods. LI-RAID composes multiple pages with different error rates into one stripe. In this process, program interference may occur between pages. LI-RAID leaves one wordline blank to mitigate the impact of program interference. These blank pages incur a small additional storage overhead (0.8% of total capacity). In the case of e-SAP, partial stripe parity overhead is added, but the parity write overhead is greatly reduced compared to the traditional RAID policy since stripes are dynamically constructed in a log-write fashion. Since these two methods are similar to the general RAID structure, other than that, the operation dependent on the method is not added in FTL and GC operations. Therefore, the proposed policy should incur overhead similar to these methods. The Differentiated Protection policy does not apply the same strength of data protection to all flash memory pages. Protection strength can be flexibly applied according to the user's request. Figure 8 shows that the difference in error rates with or without parity overhead is not distinct. WARM [6] is similar to the hot/cold-aware data placement policy. WARM dealing with retention errors minimizes additional overhead by eliminating redundant refresh operations for write-hot data. As with WARM, our hot/cold-aware data placement policy also migrates data only when data movement is required such as garbage collection or scrubbing operation. Thus, we believe that our proposed policy causes an overhead similar to WARM.

### 6. Conclusions

In this paper, we present two error handling techniques considering the characteristics of 3D-NAND flash memory. As the frequency of error occurrence varies by page and layer when ECC is used collectively, the efficiency of error correction decreases. Furthermore, one of the layers is more or less vulnerable to retention error, and this characteristic significantly affects the deterioration of the flash memory's reliability [1].

We present the number of errors after the endurance test, which shows the characteristics of different error rates for each page and layer. It shows that the ECC operation that provides the same data protection strength to the pages of the flash memory is not suitable. In addition, by classifying the data through the k-means clustering algorithm, the regions are classified according to the number of errors, and the differentiated protection strength is applied by adding parity data. The SSD model that protects data using only the existing ECC and the Differentiated Protection model we proposed are evaluated in terms of error rates. Through the distribution of the number of occurrences of retention errors, the hot data are placed to areas that are relatively more vulnerable to retention errors. Based on 9K P/E cycles, it is shown that the BER caused by retention error is lowered and the data placement structure and Differentiated Protection model presented in the paper improve the reliability of 3D-NAND flash memory through quantitative evaluation.

**Author Contributions:** Conceptualization, S.S. and J.K.; methodology, S.S. and J.K.; software, J.K.; validation, S.S. and J.K.; formal analysis, S.S. and J.K.; investigation, S.S. and J.K.; resources, S.S. and J.K.; data curation, S.S.; writing—original draft preparation, S.S.; writing—review and editing, S.S. and J.K.; visualization, S.S.; supervision, J.K.; project administration, J.K.; funding acquisition, J.K. All authors have read and agreed to the published version of the manuscript.

**Funding:** This research was funded by [National Research Foundation of Korea (NRF)] grant number [No. 2021R1F1A1063524] And The APC was funded by [Gyeonsang National University].

**Conflicts of Interest:** The authors declare no conflict of interest.

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
