# Peer review of "Differentiated Protection and Hot/Cold-Aware Data Placement Policies through k-Means Clustering Analysis for 3D-NAND SSDs"

_electronics, doi:10.3390/electronics11030398_

Round 1

Reviewer 1 Report

Please find below the comments for the manuscript by Son et al.

  1. Can the authors please provide a summary of the advantages of the present approach over the state-of-the-art?
  2. Is this study theoretical? What is the potential for its use in practice? How much it is complicated. Please provide relevant claims, justifications.
  3. Typos in the introduction, at line 44.
  4. The last paragraph of the introduction section needs re-structuring. 
  5. Most of the figures are referenced. What's the novelty. Please explain. 

Author Response

We appreciate your valuable comments. Please refer to the below answers.

  1. The advantages and differences of our study are described in comparison with the related latest studies. 
    Please refer to Sections 2.3, 2.8 of our revised manuscript. 
  2. This study is not a theoretical, but a practical suggestion. 
    The method we propose is similar to LI-RAID and WARM. 
    Also, the overhead is similar to them.
    Please refer to Sections 2.3, 2.8, 5.3 of our revised manuscript. 
  3. A typo in line 44 of the introduction has been corrected. 
    Please refer to our revised manuscript.
  4. The last paragraph of the introduction section has been reorganized into the content of how the remainder of the paper is structured. 
    Please refer to our revised manuscript.
  5. A method to utilize the physical characteristics of 3D NAND flash memory based on error rate of 3D NAND flash memory has been added to the contribution list in Section 1.  
    In addition, the differences from the latest studies related to our study are described in Sections 2.3 and 2.8.
    Please refer to our revised manuscript.

Reviewer 2 Report

Dear Authors

Congratulations, your research is well organized in terms of graphics and content. 

Since SSD devices have become one of the most used to increase the performance of computers, these also by their very structure present problems in the process of storing information.  

I suggest you explain in a better way the methodologies used, and some results about the additional time in the storage process when using your proposal. 

Author Response

We appreciate your valuable comments. Please refer to the below answers.

The additional overhead that arises when using the technique we propose in this paper is described in Section 5.3 by comparing it with the related latest studies. 
Please refer to our revised manuscript. 

Reviewer 3 Report

The authors presented a new method to improve the reliability and lifespan of 3D Flash Memory.  However, the paper has few issues:

1- Most equations are not thoroughly explained (e.g. the terms in equations 1-6 need to be described)

2- The experimental part is not explained in depth. It would be helpful to explain it in details as its the place where the authors are supposed to measure the efficacity of their approach. 

Author Response

We appreciate your valuable comments. Please refer to the below answers.

  1. Added explanations of terms in equations used in section 4 and explanations of formulas used. 
    Please refer to our revised manuscript. 
  2. Added explanations to section 5.1, such as the experimental environment and the reason for the experiment. 
    Please refer to our revised manuscript.

Reviewer 4 Report

This paper is an extension of a conference paper. The author presents two error handling techniques considering the characteristics of 3D-NAND flash memory. Comments:

1) For retention, the worst case should be considered, such as whether the error rate at high temperature is consistent with the author's analysis. Can the scheme proposed by the author work efficiently?

2) The method proposed by the author increases the design complexity, for example, the controller design is more difficult, which also needs to be analyzed and explained.

Author Response

We appreciate your valuable comments. Please refer to the below answers.

  1. The 3D NAND flash error rate figure shown in this paper was extracted from the reference paper, and factors such as temperature and voltage that can affect the error rate measurement were also referred to the reference paper.
  2. The methods we proposed in this paper are similar to LI-RAID and WARM.
    Also, the overhead is similar to them, so there is no significant additional overhead in designing and implementing the controller. 
    Please refer to Section 5.3 of our revised manuscript.

Round 2

Reviewer 2 Report

Dear Authors

It is much better.